# The Role of GDF15 as a Myomitokine

**DOI:** 10.3390/cells10112990

**Published:** 2021-11-03

**Authors:** Kornelia Johann, Maximilian Kleinert, Susanne Klaus

**Affiliations:** 1Group of Muscle Physiology and Metabolism, German Institute of Human Nutrition, Potsdam-Rehbruecke (DIfE), 14558 Nuthetal, Germany; kornelia.johann@dife.de (K.J.); Maximilian.Kleinert@dife.de (M.K.); 2Section of Molecular Physiology, Department of Nutrition, Exercise and Sports, Faculty of Science, University of Copenhagen, 2200 Copenhagen, Denmark; 3Department of Physiology of Energy Metabolism, German Institute of Human Nutrition, Potsdam-Rehbruecke (DIfE), 14558 Nuthetal, Germany; 4Institute of Nutritional Science, University of Potsdam, 14469 Potsdam, Germany

**Keywords:** anorexia, appetite regulation, cardiokine, cytokine, exercise, mitochondria, muscle, myokine, myopathy, sarcopenia

## Abstract

Growth differentiation factor 15 (GDF15) is a cytokine best known for affecting systemic energy metabolism through its anorectic action. GDF15 expression and secretion from various organs and tissues is induced in different physiological and pathophysiological states, often linked to mitochondrial stress, leading to highly variable circulating GDF15 levels. In skeletal muscle and the heart, the basal expression of GDF15 is very low compared to other organs, but GDF15 expression and secretion can be induced in various stress conditions, such as intense exercise and acute myocardial infarction, respectively. GDF15 is thus considered as a myokine and cardiokine. GFRAL, the exclusive receptor for GDF15, is expressed in hindbrain neurons and activation of the GDF15–GFRAL pathway is linked to an increased sympathetic outflow and possibly an activation of the hypothalamic-pituitary-adrenal (HPA) stress axis. There is also evidence for peripheral, direct effects of GDF15 on adipose tissue lipolysis and possible autocrine cardiac effects. Metabolic and behavioral outcomes of GDF15 signaling can be beneficial or detrimental, likely depending on the magnitude and duration of the GDF15 signal. This is especially apparent for GDF15 production in muscle, which can be induced both by exercise and by muscle disease states such as sarcopenia and mitochondrial myopathy.

## 1. Introduction

Growth differentiation factor 15 (GDF15) has been identified over 20 years ago as a cytokine in different patho-physiolgical contexts owing to the fact that it can be expressed and secreted by multiple tissues and cell types in response to cellular stress. Only in recent years has it been implicated as an important factor in the regulation of metabolic health and energy metabolism. More specifically, its role in appetite regulation has been emphasized since the discovery of the glial cell line-derived neurotrophic factor (GDNF) receptor alpha-like (GFRAL) as its hindbrain located receptor, which mediates the anorectic effects of GDF15. The general biology of GDF15 and its role in health and disease has been addressed in excellent and thorough recent reviews [1,2]. Here, we will focus on the role of GDF15 as a mitokine, which is induced by mitochondrial stress and dysfunction as a possible mitohormesis mediator [3]. In particular, we will address the still controversial issue of GDF15 as a myomitokine [4] and its possible role in muscle-brain crosstalk.

## 2. Mitohormesis Concept

The hormesis theory describes the triggering of beneficial health effects as a response to repeated stresses of mild intensities [5]. Accordingly, the concept of mitohormesis (mitochondrial hormesis) has been proposed as an adaptive stress response induced by repeated or chronic mild perturbations of mitochondrial function by diverse stressors. It is now well accepted that mitochondrial stress can lead to alterations in cytosolic and nuclear signaling, which induces cytoprotective pathways, leading to an increased stress resistance. Mitohormetic pathways can act in a cell-autonomous fashion to preserve cellular function and survival but they can also affect and improve systemic metabolism potentially leading to an improvement of health and even an increased lifespan [3,6,7,8,9]. For example, mitohormesis can be triggered by a mild mitochondrial uncoupling via chemical uncouplers or uncoupling proteins, which was shown to improve metabolic health, thereby affecting aging and longevity [3,10].

Parts of the systemic mitohormesis effects are thought to be mediated by mitokines, that is molecules such as peptides or cytokines released by cells in response to mitochondrial stress, which act on other cells or tissues. In the narrow, original sense of meaning as first proposed in 2011 [11], mitokines were defined as mitochondrial-derived peptides (MDPs). Today there are eight known MDPs, including MOTS-c (encoded by the 12S ribosomal RNA gene), as well as humanin and six small humanin-like peptides (SHLP1–6), which are encoded by the 16S ribosomal RNA gene. All of these MDPs have been demonstrated to have different cell protective or systemic beneficial properties [12].

Later on, the idea of mitokines was extended to include not only mitochondria-coded peptides but also cytokines encoded by nuclear genes, which are induced by mitochondrial dysfunction and act as endocrine or paracrine factors. This broader definition was first applied to fibroblast growth factor 21 (FGF21), which was found to be induced in mouse models of autophagy deficiency by targeted disruption of Atg7 (autophagy-related 7) in skeletal muscles or the liver, leading to a protection from obesity and insulin resistance of these mice [13]. More recently, GDF15 has come into the focus as an additional mitokine, and today FGF21 and GDF15 are considered as two important, endocrine-acting mitokines that are able to initiate systemic adaptations affecting overall energy and substrate metabolism [3,8,14].

## 3. GDF15 as a Myokine—Signaling and Physiological Significance

### 3.1. GDF15—A Pleiotropic Biomarker of Various Physiological Stress Situations

GDF15 was first identified in 1997 and described as a member of the transforming growth factor β (TGFβ) superfamily, which was found to be secreted as a dimeric protein by activated macrophages, and reported as a human cDNA with transcripts highly expressed in the placenta [15,16]. It was initially named MIC-1 (macrophage inhibitory cytokine-1) and also given various other names such as placental bone morphogenetic protein (PLAB), placental transforming growth factor-β (PTGF-β), prostate-derived factor (PDF), and non-steroidal anti-inflammatory drugs-activated gene (NAG1) [1,17], depending on its site of expression and perceived function. Today, GDF15 is regarded as a member of the GDNF family, a divergent group within the TGF-β superfamily, rather than a classical GDF family member [1,18]. GDF15 is synthesized as a monomeric precursor protein of around 40kDa that is cleaved after dimerization to yield the mature dimer of ~30 kDA, which is subsequently secreted by a still unknown mechanism [19].

In normo-physiological conditions, GDF15 mRNA is expressed by a wide variety of cells and tissues such as the kidney, lung, pancreas, heart, skeletal muscle, adipose tissue, liver, gastrointestinal tract, and brain, with the highest expression in placenta. In humans, GDF15 protein expression is high in placenta, medium in prostate and urinary bladder, and low in gastro-intestinal tract, pancreas, and kidney, according to the human protein atlas [20]. Interestingly, basal circulating GDF15 levels of 0.2–1.2 ng/mL in humans are rather high compared to other cytokines [1,18]. Circulating GDF15 increases with aging and today GDF15 is recognized as a reliable biomarker of aging [21], reaching levels of around 3–5 ng/mL in centenarians [22]. In line with its high expression in placenta, circulating GDF15 is very strongly increased during pregnancy, reaching levels as high as 60–70 ng/mL in the third trimester [23]. Furthermore, GDF15 is increased in a number of different pathologies and diseases (many of those age related) such as chronic inflammatory disease, cardiovascular and renal disease, as well as different malignancies. Circulating GDF15 concentrations in advanced cancers are in the range of 10–100 ng/mL. GDF15 can thus be considered as a general marker of disease, which is increasing with advanced disease and associated with all-cause mortality (reviewed by [1]). Also in the very old, GDF15 is a predictor of mortality, showing an especially strong increase in centenarians [22]. The secretion of GDF15 from various malignant tumors and its mediation of tumor-associated cachexia and weight loss was established in 2007 [24] and even earlier (in 2000) it was shown that the placental trophoblast is a major source of GDF15 in maternal plasma and amniotic fluid [25]. GDF15 has been linked to the induction of nausea and vomiting in pregnancy, possibly as a side effect. Although it is thought to play an important physiological role in human pregnancy, the mechanism remains enigmatic (reviewed in [1,26]). Another non-pathological stress situation, which leads to an increase of circulating GDF15, is physical exercise. For example, in elite male triathletes. plasma GDF15 was about 4-fold increased after 4 h of vigorous bicycling [27].

While the tissue source of GDF15 is quite obvious in cancer and pregnancy, it is less obvious for exercise. Vigorous exercise acutely induces GDF15 gene expression in skeletal muscle, but also in liver and heart [27,28], and basically nothing is known about the relative contribution of different tissues or organs to circulating GDF15 in normo-physiological conditions. This obviously makes its use as a specific biomarker difficult, because it is not possible to relate plasma GDF15 concentrations to specific pathologies. GDF15 gene expression can be upregulated in different situations of cellular stress caused by e.g., inflammation, infection, neoplasia, ionizing irradiation as well as different drugs via various transcription factors such as p53, kruppel-like factor-4 (KLF4), early growth response protein 1 (EGR1), hypoxia-inducible factor-1α (HIF-1α), activating transcription factor 3 and 4 (ATF3, ATF4), and C/EBP homologous protein (CHOP) [18,26,29].

As crucial organelles for cellular energy generation, biosynthetic pathways, and cellular signaling, mitochondria and their functions are affected by a whole range of situations leading to cellular stress and injuries, as observed in disease processes, environmental exposures, and aging. Not surprisingly, therefore, mitochondrial dysfunction is the hallmark of a wide range of pathologies and metabolic disorders and has been suggested as a common underlying factor responsible for the induction of GDF15 in these highly variable pathophysiologic contexts [29,30]. Cellular stress such as mitochondrial stress activates the integrated stress response (ISR), a common adaptive pathway for the restoration of cellular homeostasis. Activation of the ISR converges on the increased translation of proteins implicated in stress recovery, such as ATF4, the main regulator of the mitochondrial stress response in mammals, which is crucial for triggering the expression of the two mitokines FGF21 and GDF15 [14]. Indeed, GDF15 can be considered as an endocrine signal of the ISR, and CHOP was demonstrated as the terminal effector for inducing GDF15 gene expression in response to activation of the ISR [4,31,32].

### 3.2. The GDF15-GFRAL Pathway and its Metabolic Role

In 2017, 20 years after the initial cloning of GDF15, GFRAL was described as the unique receptor for GDF15 independently by different groups [33,34,35,36]. They showed that GFRAL, an orphan receptor of the GDNF receptor α family, signals through the tyrosine kinase co-receptor Ret and demonstrated a GFRAL dependent reduction of food intake and body weight after pharmacological application of GDF15. Interestingly, GFRAL expression was found to be specific to the hindbrain (area postrema, AP, and nucleus of the tractus solitarius, NTS). The AP is a brain region of special interest regarding brain-periphery communication because of its location outside the blood–brain barrier. As a circumventricular organ, the AP is able to integrate a large amount of signal from the periphery due to a high vascularization in this region [37].

Today, the best-explored metabolic role of the GDF15–GFRAL pathway is the control of food intake, implying a role in the regulation of energy metabolism (Figure 1). GDF15 over-expressing mice are resistant to the development of diet-induced obesity and associated metabolic disorders (see [3] for a review). On the other hand, the GDF15-GFRAL pathway does not seem to be implicated in appetite control in normo-physiological situations. Neither GDF15 nor GFRAL ablated mice show major alterations in body weight or food intake and preference (reviewed in [2]). Concerning its role in pathophysiology, GDF15 has been linked to cancer cachexia and side-effects of chemotherapy. Inhibition of GDF15 activity with antibodies targeting either GDF15 or its receptor GFRAL was found to reverse chemotherapy-induced anorexia and emesis and cancer cachexia in mice and nonhuman primates [38,39].

There is still considerable lack of knowledge and ongoing controversies about the physiological importance of the GDF15–GFRAL pathway. For example, while activation of the GDF15–GFRAL pathway can lead to cancer-associated anorexia-cachexia syndrome, it can also ameliorate obesity-associated morbidities. Some of these discrepancies could depend on the physiological context and the duration and magnitude of the GDF15 signal, which has important therapeutic implications for approaches targeting the GDF15–GFRAL axis.

Recent research on the GDF15–GFRAL axis has focused on its anorectic action and suggests that it acts as a mediator of the physiological responses to visceral malaise states such as the induction of food aversion, nausea, and emesis preceding the onset of anorexia, as shown in different rodent models and shrews [31,40,41,42]. This fits with the well-known actions of the hindbrain AP in the control of nausea and vomiting [43] beside its role as a control center of food intake [44].

It was already shown in 2007 that GDF15 injection in mice leads to activation of neurons in hindbrain as well as in hypothalamic areas involved in appetite regulation such as the paraventricular nucleus of the hypothalamus (PVH) [24] but the nature of the GFRAL-positive neurons in the hindbrain and the central circuitry leading to an inhibition of food intake upon GFRAL stimulation is just starting to emerge. One study suggested that the primary target for GDF15 is a distinct population of GFRAL/CCK neurons to engage the neural circuitry involved in anorexia and conditioned taste aversion [45]. In another approach, single-nucleus RNA-seq was employed to create an atlas of AP cell types that mediate nausea-associated behaviors. One of four identified excitatory neuronal clusters was characterized by the expression of GFRAL, and specific ablation of these GFRAL neurons abolished both lithium chloride- and lipopolysaccharide-conditioned taste aversion in mice, suggesting that this cell cluster is mediating behavioral responses to different visceral malaise-inducing stimuli [46]. GFRAL neurons most strongly innervate the parabrachial nucleus (PBN) but seem to have no direct communication with critical hypothalamic sites associated with the long-term regulation of body weight. Furthermore, silencing of GFRAL target neurons in the PBN abrogated the aversive and anorexic effects of GDF15 [47]. As an interface between brain stem and forebrain, the PBN is involved in the relay of visceral sensations to the hypothalamus and cortico-limbic structures such as the central nucleus of the amygdala (CeA), which is also a downstream target of GDF15 and GFRAL neurons [33,47].

Importantly, evidence for the induction of malaise states and food aversion through the GDF15–GFRAL pathway was obtained using supraphysiological doses of exogenously administered GDF15 or acute stimulation of the GDF15–GFRAL pathway. It is not known if anorexia induced by endogenously elevated GDF15 is also linked to malaise states. GFRAL-expressing neurons co-express the glucagon-like peptide 1 receptor (GLP1R) [46]. GLP1R agonists are used as anti-diabetic drugs in humans where the most common side-effect is the induction of nausea [48]. Therefore, it could be speculated that the over-activation of GFRAL/GLP1R neurons by supraphysiological doses of GDF15 lead to nausea as a secondary side-effect.

In this context, it is of interest that a comprehensive behavioral characterization revealed a decreased anxiety behavior together with an increased exploratory behavior in GDF15^−/−^ mice [49]. This suggests that GDF15 might be effective to induce stress and anxiety. This fits with recent observations in rodent models that exogenous GDF15 application as well as activation of endogenous GDF15 by genotoxic or endoplasmic reticulum toxins led to an activation of the hypothalamic–pituitary–adrenal (HPA) stress axis, which was absent in GDF15^−/−^ [50]. In a mouse model of GDF15 induction as a myomitokine, we could show that endogenous activation of the GDF15–GFRAL pathway leads to increased anxiety related behavior and induces a day-time restricted anorexia by increased hypothalamic corticotropin releasing hormone (CRH) acting via CRH receptor 1 [51].

Interestingly, the body weight lowering effect of GDF15 could be dissociated from its anorectic effect in a cancer model characterized by high GDF15 levels. GDF15 has been linked to cancer cachexia and inhibition of GDF15–GFRAL activity by a GFRAL targeting antibody reversed cancer cachexia in mice [39]. Mechanistic exploration suggested that GDF15 induced a lipolytic response in adipose tissue, independently of anorexia, mediated by the peripheral sympathetic nervous system (SNS) [38].

### 3.3. Metabolic Role of the Skeletal Muscle

The skeletal muscle is a dynamic organ capable of adaptive remodeling in response to aging, starvation, metabolic disorders, and physical exercise [52]. One can picture the build of an elite endurance runner and a body builder to appreciate the spectrum of plasticity of skeletal muscle. On average, the skeletal muscle in humans contributes 40% and 30% to male and female body mass in nonobese humans. The musculature stores 300–500 g of glucose in the form of glycogen [53], which, for most people, represents in theory sufficient energy to complete a half marathon. At rest, the skeletal muscle accounts for ~20% of the body’s metabolic rate in normal-weight adults [54]. During strenuous exercise, however, the metabolic rate of contracting the skeletal muscle can increase >20-fold. During maximal exercise efforts, muscle energy turnover accounts for up to 90% of the overall metabolic rate, highlighting an enormous metabolic flexibility. Muscles can use different energy substrates, such as glucose, fatty acids and ketone bodies, to produce energy in the form of ATP. During high-intensity exercise, glucose is the preferred energy substrate and muscle glucose uptake from the circulation can increase more than 100-fold compared to rest [55]. Conversely, during prolonged periods of fasting, skeletal muscle can trigger catabolic reactions to break down glycogen, fats, and proteins to supply energy or energy precursors to other organs, most importantly, the brain. 

Skeletal muscle is a heterogeneous tissue composed of type 1, 2a, 2x (2b in rodents) fibers as well as other cell types (e.g., satellite cells). Broadly speaking, slow-twitch type 1 fibers are smaller, less powerful, but more fatigue-resistant. In contrast, fast-twitch type 2a and 2x fibers are bigger, more powerful but also fatigue quicker. Type 2 fibers are specialized to rapidly produce ATP from creatine phosphate and anaerobic glycolysis to sustain brief and explosive movements like sprinting. Type 1 fibers, on the other hand, have more mitochondria and rely heavily on oxidative phosphorylation to meet the energy demands of longer bouts of lower-intensity movement. Strenuous exercise poses a transient physiological stress for muscle. For example, mechanical stress, especially during loaded eccentric contractions, strains or even damages muscle fibers. Rapid production of ATP during intense exercise via anaerobic glycolysis leads to buildup of lactate and to changes in pH. During most exercises, the majority of ATP is produced by mitochondria from oxidative phosphorylation. This can generate reactive oxygen species (ROS) such as superoxide and hydrogen peroxide and the subsequent formation of free radicals that can damage cells. This multitude of stressors triggers the activation of specific signaling pathways, controlling exercise-induced gene expression and protein synthesis in the contracting muscles [56]. This includes the induction of endoplasmic reticulum (ER) stress and the unfolded protein response (UPR) pathways, which play chief roles leading to general improvement of skeletal muscle mass and function. In mice, it has been shown that the UPR mediates skeletal muscle adaptation to exercise [57] and repeated activation of the UPR by exercise is an important contributor to exercise-induced skeletal muscle remodeling [58]. Skeletal muscle adaptations to exercise training are the result of repeated, transient bursts in mRNA expression during each exercise bout that have an aggregate effect and lead to increases in transcription of nuclear- and mitochondrial-encoded proteins [59]. Collectively, these adaptations fortify skeletal muscle to better cope with the stress imposed by subsequent exercise.

It is well accepted that skeletal muscle is a secretory organ that releases so-called myokines. These muscle-derived proteins are released into the circulation to act in an auto/paracrine or endocrine manner [60]. Indeed, muscle cells have the capacity to secrete hundreds of factors [61]. Exercise is a potent stimulus for the release of myokines from contracting muscle. This includes many mitokines, which are triggered in response to mitochondrial stress imposed during contractions and that contribute to the multitude of metabolic and physiological changes associated with exercise [61].

### 3.4. GDF15 as a Myokine and Cardiokine

Circulating GDF15 levels are elevated in a number of physiological and pathophysiological states and have been implicated as a biomarker for severity of cardiovascular disease (reviewed in [62]). Furthermore, GDF15 has also been implicated in exercise and exercise recovery, as described in further detail below. Gene expression of Gdf15 has been shown to be regulated by p53 [63], as well as CHOP [4,31,32]. While p53-mediated transcription of Gdf15 is induced in response to several stressors, such as hypoxia and inflammation [62,64], CHOP is induced through the mitochondrial unfolded protein response (UPRmt) and the final effector of the ISR, as already mentioned. In cardiomyocytes, Gdf15 expression seems to be induced by nitric oxide signaling, as well as IL1-β/IFN-γ [65]. Muscle mitochondrial stress can be induced by a variety of factors such as increased mitochondrial respiration, elevated ROS production, or dysfunction of oxidative phosphorylation (Figure 2). Several mouse models of muscle mitochondrial stress were shown to have increased GDF15 expression in muscle, as well as increased circulating GDF15 levels [4,66,67] (Table 1). In a mouse model with a muscle-specific Crif1 deficiency, muscle-derived GDF15 was shown to be induced through the UPRmt-ATF4-CHOP axis. These mice were resistant against diet-induced obesity and insulin resistance, due to an increase in lipolysis in adipocytes and hepatocytes, which was shown to be GDF15-mediated [4]. It remains to be elucidated how GDF15 signaling in adipocytes and hepatocytes is mediated, since its receptor GFRAL has been shown to be expressed specifically in the hindbrain, where it mediates the effects on appetite, hunger, and satiety, as described above [33,34,35,36]. In another mouse model of mitochondrial dysfunction caused by Ant1-deficiency which leads to an excess of ROS production, Fgf21 and Gdf15 gene expression was highly induced in muscle [67].

One of the best studied mouse models of mitohormesis and mitokine expression is the UCP1-tg mouse with skeletal muscle directed, low expression level of the uncoupling protein UCP1, leading to a slightly compromised skeletal muscle mitochondrial function due to increased uncoupling of the respiratory chain [68,69,70]. Muscle-targeted respiratory uncoupling has been shown to increase longevity and promote healthy aging in mice [71,72]. The healthy metabolic phenotype of UCP1-tg-mice is characterized by increased energy expenditure, delayed diet-induced obesity development, reduced hepatic steatosis, browning of white adipose tissue (WAT), and improved glucose homeostasis, despite a decreased muscle mass and strength [71,73,74,75]. The UCP1-dependent activation of the ISR in the skeletal muscle leads to the induction of Fgf21 and Gdf15 gene expression, resulting in an over 5-fold increase in their circulating levels in UCP1-tg mice. Generation of either Fgf21 or Gdf15 ablated UCP1-tg mice showed that FGF21 was of minor importance for the metabolic adaptations of this particular mouse model [76], whereas ablation of Gdf15 demonstrated its crucial impact on the metabolic phenotype of UCP1-tg mice. Loss of GDF15 led to a progressive body mass increase, which was exclusively due to an accumulation of body fat, while lean body mass was not affected. Further inquiry into the metabolic adaptations of UCP1-tg mice showed that their WAT remodeling (browning) as well as their increased insulin sensitivity were abolished upon loss of GDF15, interestingly despite increased levels of circulating FGF21 [66]. This first appears puzzling because FGF21 is known to induce WAT browning [77] and the most striking effect of FGF21 ablation in UCP1-tg mice was a complete loss of browning and WAT remodeling [76]. However, recently it was shown that the induction of browning by ß-adrenergic stimulation requires an autocrine action of FGF21 while circulating FGF21 is dispensable for WAT browning [78]. GDF15 was shown to increase sympathetic outflow to adipose tissue [38], suggesting that the increased WAT browning of UCP1-tg mice was due to a GDF15-induced adrenergic stimulation of FGF21 production in adipose tissue acting in an auto/paracrine fashion to induce the expression of thermogenic genes. This auto/paracrine induction of browning was then prevented by the whole body knock out of FGF21. Of note, in UCP1-tg mice, there were no apparent effects of either FGF21 or GDF15 on skeletal muscle morphology or function itself, suggesting their predominant endocrine action when induced as mitokines.

Interestingly, GDF15 expression in the heart has also been implicated in a variety of disease states. Pediatric heart disease is associated with a failure to thrive, which is characterized by impaired physical growth and associated with decreased circulating insulin growth factor 1 (IGF1) levels [79,80]. Recently, heart-derived GDF15 has been shown to act directly on hepatocytes to inhibit growth hormone signaling through a mechanism involving decreased STAT5 phosphorylation [81]. Furthermore, cardiac GDF15 also seems to have protective autocrine/paracrine effects in models of cardiac hypertrophy [82], as well as ischemia/reperfusion injury in a PI3K/Akt-dependent manner [65,83]. This is supported by patient data that exhibit enhanced myocardial GDF5 pro-peptide expression in acute myocardial infarction [65].

### 3.5. GDF15 in Exercise versus Sarcopenia

Exercise is a potent physiological stimulus for glucose uptake in skeletal muscle and exercise improves their insulin sensitivity for up to two days following the exercise bout. As such, exercise is one of the best therapeutic options to treat or prevent insulin resistance and type 2 diabetes [87]. We and others have shown that aerobic exercise as well as resistance exercise increase circulating GDF15 in humans [27,88,89,90,91,92]. Notably, GDF15 increases during and also in recovery from exercise. For example, following 1 h of vigorous cycling, circulating GDF15 increased from 0.22 to 0.3 ng/mL immediately after exercise and further increased to 0.35 ng/mL 3 h into recovery from exercise [90]. Within 24 h, exercise-induced increases in GDF15 return back to baseline levels [27]. Although the relationship between exercise duration, intensity, and GDF15 levels has not been rigorously tested, it appears that high-intensity exercise lasting for more than 2 h can elicit increases in circulating GDF15 to concentrations comparable to some disease states. For example, after 4 h of intense cycling, GDF15 increased from baseline levels of ~0.3 ng/mL to ~2 ng/mL in elite male triathletes [27]. Similarly, circulating GDF15 increased from ~0.5 ng/mL to ~1.6 ng/mL after a 130 km long cycling race in a study cohort of mostly middle-aged men [89] or from ~0.35 ng/mL to ~1 ng/mL in middle-aged male runners after completing a marathon in 3–4 h [88].

Which tissue GDF15 is secreted from during exercise remains to be determined. In mice, Gdf15 mRNA is increased in the liver, heart, and soleus muscle after treadmill running, which indicates that these organs secrete GDF15 during exercise [27,28]. Three hours of electric stimulation-induced contractions of primary human muscle cells triggered the release of GDF15 into media [91] and Gdf15 mRNA and protein secretion were induced in C2C12 myotubes by the mitochondrial stressor oligomycin [93], demonstrating that muscle cells have the capacity to secrete GDF15 in response to being metabolically challenged. In humans, however, GDF15 levels were similar in plasma sampled from the femoral artery and femoral vein before, during, and 1 h after vigorous cycling exercise, indicating that the exercise-induced rise in circulating GDF15 occurs without direct contribution from a contracting skeletal muscle [90]. A possibly contribution of skeletal muscle during longer exercise bouts, when GDF15 levels increase to greater levels, remains to be investigated. It also remains possible that muscles secrete GDF15 during exercise, which then acts locally. For example, it has been proposed that GDF15 can act inside cells to modulate transcriptional regulation [94]. Understanding possible GFRAL-independent effects of GDF15 biology clearly is an exciting new frontier.

GDF15 is thought to signal aversion to protect against noxious behavior [31,40]. It is therefore tempting to speculate that increases in GDF15 during exercise initiates an endocrine feedback loop that signals exercise aversion through GFRAL to prevent injurious behavior. Injection with exogenous GDF15 reduced voluntary running in wildtype mice but not in GFRAL^−/−^ mice, supporting the concept that a GDF15–GFRAL axis promotes exercise aversion [27]. As in humans, circulating GDF15 levels increase in mice following intense treadmill running, which is also associated with subsequent decreased voluntary running activity [27]. However, this blunting of voluntary running by prior exercise also occurred in GFRAL^−/−^ mice, indicating that this behavioral adjustment is not solely regulated by the endogenous GDF15–GFRAL axis [27]. It remains to be investigated whether GDF15 in concert with other factors regulates exercise motivation.

A recent publication reported that GDF15 activates the HPA axis, resulting in increased circulating glucocorticoids (corticosterone in rodents, cortisol in humans) [50]. One of the key functions of corticosterone/cortisol is to increase blood glucose levels by increasing hepatic glucose output. Circulating cortisol increases during exercise in humans [95] with more pronounced increases with longer [96] and more intense exercise bouts [95]. Whether a GDF15-GFRAL-HPA axis promotes glucose supply to contracting muscle during exercise remains to be clarified. Overall, the function of GDF15 during and after exercise is unclear and it will need to be determined if GDF15 mediates any of the many health benefits of exercise.

Contrary to the health-promoting effects of exercise, GDF15 is also elevated in diseases associated with muscle weakness such as sarcopenia and mitochondrial myopathy. It is important to consider physiology versus pathophysiology when assessing the biological consequences of elevated GDF15 levels. In contrast to the transient eustress of exercise, chronic stress due to mitochondrial dysfunction or sarcopenia is largely detrimental to muscle quality and quantity. Therefore, although GDF15 levels increase after exercise, they return to baseline levels within 8–24 h thereafter, and people that are physically fit seem to generally have lower GDF15 levels [89]. In contrast, with mitochondrial myopathy or sarcopenia, GDF15 levels remain chronically elevated, presumably resulting in vastly different biological outcomes, similar to most metabolic hormones, where chronic hypo- or hyper-physiological levels are implicated in disease states.

Sarcopenia encompasses a type of muscle atrophy that occurs with aging or immobility. It is characterized by a progressive loss of skeletal muscle mass, quality, and strength. Age-related sarcopenia is determined by a complex interplay, among many factors such as physical activity, co-morbidities, genetics, and nutrition. Sarcopenic patients with chronic obstructive pulmonary disease have GDF15 levels that are 1 ng/mL higher than those of matched controls. In these patients, circulating GDF15 also correlates inversely with quadriceps mass and exercise capacity [85]. Similarly, in older intensive-care patients with muscle atrophy, GDF15 levels averaged 7 ng/mL with some patients having levels higher than 15 ng/mL, which was seven times higher than levels in matched controls that had undergone elective surgery [97]. A recent study highlights potentially important sex differences finding that higher GDF15 concentrations were associated with lower measures of muscle mass in men but not in woman [98]. It is not clear whether compromised muscle function during aging triggers GDF15 release from muscle or whether high GDF15 contributes to loss of muscle mass. Interestingly, when Patel et al. overexpressed GDF15 in muscle, this resulted in a decrease of local muscle mass compared to contralateral sham-treated muscles, indicating that GDF15 causes atrophy [85]. Considering that mouse muscle do not express GFRAL, it is unclear how this is mediated. Like sarcopenia, mitochondrial myopathy results in loss of muscle mass, quality, and strength. While sarcopenia is a progressive, complex disease of aging, mitochondrial myopathy is triggered by inborn genetic defects that impair mitochondrial function. Patients of all ages, including children, with mitochondrial myopathy have higher levels of circulating GDF15 typically between 2–10 ng/mL [99,100,101,102] compared to matched control groups, which exhibit levels of about 0.5 ng/mL. Interestingly, inborn genetic errors unrelated to mitochondrial dysfunction that lead to myopathy are not associated with high circulating GDF15. Patients with metabolic myopathy or muscular dystrophy have GDF15 levels comparable to control subjects [99,103]. This underscores the notion that GDF15 is a mitokine. 

### 3.6. Diurnal Aspects of GDF15 Expression and Secretion

There is evidence from cell culture studies that *Gdf*15 is an oscillating, clock-controlled gene [104,105] and in humans *Gdf15* was found among genes whose expression is affected by the individual chronotype [106]. So far, there are only limited data on diurnal variations of circulating GDF15 in humans. In an Asian cohort of healthy individuals, circulating GDF15 was found to show diurnal variations with a peak around midnight and a nadir approximately at noon [107]. This pattern of diurnal variation of increased night time values was recently confirmed in young healthy male Caucasians undergoing an overfeeding regime, which did not affect the diurnal pattern [27]. However, the circulating GDF15 levels and the amplitude of their oscillations was quite low in both studies. To the best of our knowledge, diurnal/circadian variations of circulating GDF15 in humans during aging, pregnancy, or pathological conditions have not yet been investigated. 

The relevance of diurnal variations of circulating GDF15 for metabolic adaptations has been demonstrated in the afore described UCP1-tg mouse model, where GDF15 is induced as a mitokine by mitochondrial uncoupling. In UCP1-tg mice, endogenous GDF15 promotes a diurnal, daytime-restricted anorectic response controlling the systemic metabolic adaptation [66]. UCP1-tg mice show highest circulating GDF15 during the day and lower levels at night (although still several-fold increased compared to WT). Considering the reversed diurnal rhythm of night active mice compared to humans, oscillations of GDF15 therefore seem to be similar in humans and mice. In UCP1-tg mice, we could demonstrate a GDF15-dependent, diurnal anorexic response that reprograms systemic energy homeostasis and metabolic flexibility [66]. Human data on possible diurnal variations of GDF15 in different physiological and pathophysiological situations are necessary in order to evaluate the possible metabolic relevance of diurnal/circadian variations of GDF15.

## 4. Conclusions

The metabolic effects of GDF15 as an almost ubiquitously expressed cytokine have received considerable attention recently, much supported by the discovery of GFRAL as the specific receptor for GDF15, which is responsible for its central actions mainly related to appetite regulation. The cellular mechanisms leading to the induction of GDF15 expression have been well explored, and the central circuits and efferences mediating the metabolic effects of GDF15 are also starting to be unraveled. On the other hand, the physiological relevance of GDF15 induction in highly divergent physiological and pathophysiological situations is still puzzling. Especially regarding GDF15 as myomitokine, there are a number of open questions. It is not yet clear if the induction of muscle GDF15 gene expression by vigorous exercise does lead to a sustained secretion into the circulation. Autocrine or paracrine actions of muscle GDF15 as well as direct peripheral actions have been suggested and there is also the possibility of GFRAL-independent effects of GDF15, which need to be further explored. Regarding the systemic metabolic effects of GDF15, we still do not know if chronically increased GDF15 levels are more harmful than beneficial, which makes the development of GDF15-GFRAL targeting clinical approaches rather challenging.

## Figures and Tables

**Figure 1 cells-10-02990-f001:**
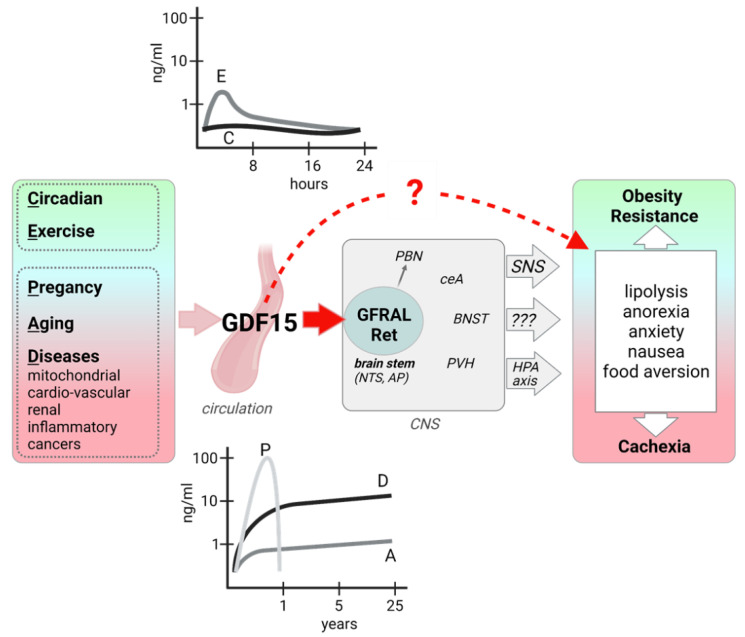
GDF15 induction, signaling, and metabolic effects. GDF15 expression and secretion from various organs and tissues is induced in different physiological and pathophysiological states, leading to highly variable circulating GDF15 levels. GFRAL, the exclusive receptor for GDF15 is expressed in neurons of the AP/NTS of the hindbrain, which directly project to the PBN and also activate other brain areas. Efferent signaling includes an increased sympathetic outflow and possibly an activation of the HPA axis. Human evidence also points to a peripheral, direct action of GDF15 on adipose tissue lipolysis. Metabolic and behavioral outcomes of GDF15 signaling can be beneficial or detrimental, possibly depending on the magnitude and duration of the GDF15 signaling. AP: area postrema; BNST: bed nucleus of the stria terminalis; ceA: central Amygdala, HPA: hypothalamic-pituitary-adrenal; NTS: nucleus tractus solitarius; PBN: parabrachial nuclei; PVH: paraventricular nucleus of the hypothalamus; SNS: sympathetic nervous system. Figure created with BioRender.com (28 September 2021).

**Figure 2 cells-10-02990-f002:**
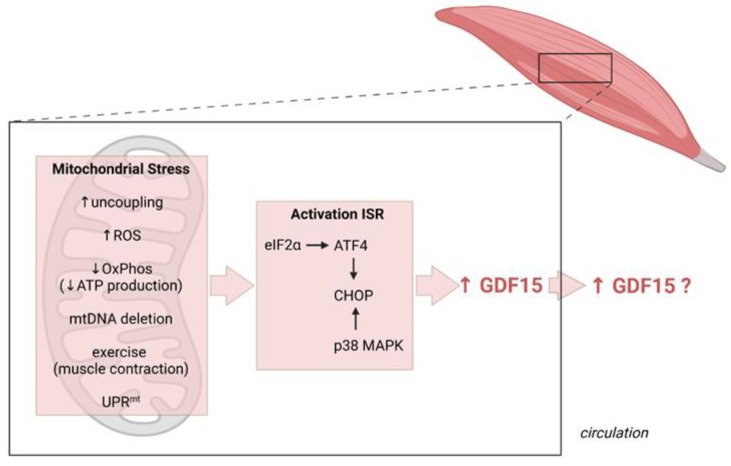
Induction of GDF15 as a myomitokine. Different (patho)physiological conditions resulting in increased muscle mitochondrial stress activate the integrated stress response (ISR) culminating in a CHOP-dependent induction of GDF15 expression which can, but does not necessarily, result in increased secretion and circulating GDF15 levels. Figure created with BioRender.com (28 September 2021).

**Table 1 cells-10-02990-t001:** Animal models implicating GDF15 as a myokine/cardiokine.

Disease Model	Pathology/Disease	Study	Main Outcome
Monocrotaline (MCT) rat	Pulmonary arterial hypertension (PAH)	[84]	GDF15-mediated phosphorylation of TAK1 leads to muscle loss in PAH
GDF15-KO mouse	Ischemia/reperfusion (I/R) injury	[65]	GDF15 protects from I/R injury
Transient GDF15 overexpression in tibialis anterior muscle (mouse)	Chronic obstructive pulmonary disease (COPD)	[85]	GDF15 contributes to muscle loss in COPD
Cardiac-specific GDF15 overexpression (mouse)	Cardiac hypertrophy	[82]	GDF15 as an autocrine/endocrine factor antagonizing hypertrophic response and loss of ventricular performance
Deletor mouse	Mitochondrial myopathy	[86]	Mitochondrial myopathy is associated with induction of ISRmt and Gdf15 mRNA in muscle
αKOγKO mouse (genotype: ERRα^−/−^ERRγ^flox/flox^Myh6-Cre^+^	Pediatric heart disease with failure to thrive	[81]	Increased cardiac-derived circulating GDF15 blocks hepatic growth hormone signaling
hGDF15 overexpressing mouse	Ischemia/reperfusion (I/R) injury	[83]	GDF15 protects from I/R injury in heart transplantation
Crif-mKO mouse	Mitochondrial dysfunction	[4]	GDF15 as a myomitokine, protects from development of obesity and insulin resistance
Ant1-KO mouse	Increased mitochondrial metabolism	[67]	ROS overproduction increases Gdf15 gene expression in muscle and prevents diet-induced obesity and insulin resistance
Ucp1-tg mouse	Mild skeletal muscle mitochondrial dysfunction	[66]	GDF15 as a mitomyokine mediates diurnal anorexia and beneficial metabolic adaptations

## Data Availability

No original data reported.

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
