# Peer review of "The Role of GDF15 as a Myomitokine"

_cells, 2021, doi:10.3390/cells10112990_

Round 1

Reviewer 1 Report

The article is well-written and provides a comprehensive view on the role of GDF 15. 

Minor comments:

In the abstract section, full form of 'HPA' should be used. 

Author Response

We would like to thank both reviewers for their efforts and their positive evaluation of our manuscript

Reviewer 1:

The article is well-written and provides a comprehensive view on the role of GDF 15. 

Minor comments:

In the abstract section, full form of 'HPA' should be used.

Answer: this was corrected.

Reviewer 2 Report

It is an interesting review article, but there are many typos/mistakes such as follows to be corrected. Although some issues depend on the journal's guideline, please revise them accordingly at first. 

L36: GDNF => Please define together with the full spelling at the first appearance like glial cell line–derived neurotrophic factor (GDNF) in the line 85. Generally, the abbreviations should be used on and after the second appearance. This rule should be also true to the other abbreviations. Please check throughout the text. 

L56: Part of the systemic mitohormesis effects are => Parts of the systemic mitohormesis effects are

L64: mitochondria coded peptides => mitochondria-coded peptides

L82: prostate derived factor => prostate-derived factor 

L125: HIF1α => HIF-1α

L145: the glial-derived neurotrophic factor (GDNF) have already been defined earlier, and please use the abbreviation only.

L147: GFRAL dependent reduction => GFRAL-dependent reduction

L176: PBN should be defined at the footnote as well.

L193: paraventricular nucleus (PVH) => paraventricular nucleus of the 
hypothalamus (PVH)

L225: GDF15-ko mice => GDF15-knock out (KO) mice, as shown in the Table 1.

L241: The musculature stores 300-500 g of glucose in the form of glycogen [54], which, for most people, represents sufficient energy to complete a half marathon.  => Is it really true? Please add the description and/or references to validate this finding.

L261: lower intensity movement => lower-intensity movement

L300: figure => Figure 

L301: GDF15 expression  =>  Gdf15 expression? In any case, please be consistent or differentiate appropriately between GDF and Gdf throughout the manuscript.

L302: table => Table

Table 1: GDf15 => GDF15 or Gdf15

Table 1: Mitochondrial Myopathy; Increased Mitochondrial 
Metabolism => Are the capitalizations needed? In any case, please be consistent within the tables.

L326: FGF21 was of minor importance => Is it true? Please refer to Rang, Y.; et al. Low-Protein High-Fat Diet Leads to Loss of Body Weight and White Adipose Tissue Weight via Enhancing Energy Expenditure in Mice. Metabolites 2021, 11, 301.

L377, 397: 1-hour of vigorous cycling => 1 hour of vigorous cycling

L378: 0,22 to 0,3 => 0.22 to 0.3 

L389: What tissue => Which tissue

L500: GFRAL independent effects => GFRAL-independent effects

There might be other potential concerns, but please check carefully throughout the manuscript once again before re-submission.

Author Response

We would like to thank both reviewers for their efforts and their positive evaluation of our manuscript

L36: GDNF => Please define together with the full spelling at the first appearance like glial cell line–derived neurotrophic factor (GDNF) in the line 85. Generally, the abbreviations should be used on and after the second appearance. This rule should be also true to the other abbreviations. Please check throughout the text. 

Answer: thank you for pointing this out, it was corrected. We also carefully checked the use of abbreviations and throughout the text and changed when necessary.

L56: Part of the systemic mitohormesis effects are => Parts of the systemic mitohormesis effects are

L64: mitochondria coded peptides => mitochondria-coded peptides

L82: prostate derived factor => prostate-derived factor 

L125: HIF1α => HIF-1α

L145: the glial-derived neurotrophic factor (GDNF) have already been defined earlier, and please use the abbreviation only.

L147: GFRAL dependent reduction => GFRAL-dependent reduction

L176: PBN should be defined at the footnote as well.

L193: paraventricular nucleus (PVH) => paraventricular nucleus of the 
hypothalamus (PVH)

L225: GDF15-ko mice => GDF15-knock out (KO) mice, as shown in the Table 1.

L261: lower intensity movement => lower-intensity movement

L300: figure => Figure 

L301: GDF15 expression  =>  Gdf15 expression? In any case, please be consistent or differentiate appropriately between GDF and Gdf throughout the manuscript.

L302: table => Table

Table 1: GDf15 => GDF15 or Gdf15

Table 1: Mitochondrial Myopathy; Increased Mitochondrial 
Metabolism => Are the capitalizations needed? In any case, please be consistent within the tables.

L377, 397: 1-hour of vigorous cycling => 1 hour of vigorous cycling

L378: 0,22 to 0,3 => 0.22 to 0.3 

L389: What tissue => Which tissue

L500: GFRAL independent effects => GFRAL-independent effects

There might be other potential concerns, but please check carefully throughout the manuscript once again before re-submission.

Answer: thank you for your careful reading. All above listed typos were corrected, we also checked and corrected inconsistencies. We carefully went over the text again and hopefully found and eliminated all additional typos and mistakes.

L241: The musculature stores 300-500 g of glucose in the form of glycogen [54], which, for most people, represents sufficient energy to complete a half marathon.  => Is it really true? Please add the description and/or references to validate this finding.

Answer: This statement was meant to illustrate the amount of energy stored in muscle as glycogen and may have been a bit superficial. The energy cost of running, however, has been assessed by many groups. One of the first seminal papers on this topic concluded that the cost of running is about 1 kcal/kg body weight per km (https://doi.org/10.1152/jappl.1963.18.2.367). Therefore, a 75 kg person burns 75 kcal per km of running. A half marathon is 21 km long. Ergo 1575 kcal would have been burned upon completion. In support, the metabolic cost of marathon running (42 km) were estimated to be ~2700 and ~2400 kcal for man and women, respectively (PMID: 18076277). Given that 1 g of glycogen should yield 4 kcal in energy, 300 – 500 g comes out to 1200 – 2000 kcal. This discussion is however beyond the scope of this review so we suggest to make the above statement a bit more speculative by adding “in theory” in the sentence. We hope the reviewer agrees with this solution.

L326: FGF21 was of minor importance => Is it true? Please refer to Rang, Y.; et al. Low-Protein High-Fat Diet Leads to Loss of Body Weight and White Adipose Tissue Weight via Enhancing Energy Expenditure in Mice. Metabolites 2021, 11, 301.

Answer: We are sorry for the misunderstanding. Of course FGF21 is an important metabolic regulator. The “minor importance” is just referring to the UCP1-tg mouse model, where we found that FGF21 ablation did not affect the overall metabolic phenotype of this model. We changed the sentence to make this more clear: “Generation of either Fgf21 or Gdf15 ablated UCP1-tg mice showed that FGF21 was of minor importance for the metabolic adaptations of this particular mouse model [77], whereas ablation of Gdf15 demonstrated its crucial impact on the metabolic phenotype of UCP1-tg mice.”

Round 2

Reviewer 2 Report

The manuscript was well corrected.